# Agomelatine prevented depression in the chronic restraint stress model through enhanced catalase activity and halted oxidative stress

Jiaxi Xu[1], Cheng Zhu[2,3], Piaopiao Jin[4], Wangdi Sun[5], Enyan Yu[6,7]*

1 Department of Psychiatry, Tongde Hospital of Zhejiang Province, Hangzhou, Zhejiang, China, 2 Kangning Hospital attached to Wenzhou Medical University, Wenzhou, Zhejiang, China, 3 School of Mental Health, Wenzhou Medical University, Wenzhou, Zhejiang, China, 4 Department of Psychiatry, Yiwu Central Hospital, Jin Hua, Zhejiang, China, 5 Department of Psychiatry, Zhejiang Hospital, Hangzhou, Zhejiang, China, 6 Department of Clinical Psychology, Zhejiang Provincial People's Hospital, Hangzhou, Zhejiang, China, 7 Department of Clinical Psychology, Cancer Hospital of the University of Chinese Academy of Sciences (Zhejiang Cancer Hospital), Hangzhou, Zhejiang, China

* yuenyan@aliyun.com

**Data Availability Statement:** All relevant data are within the paper and its Supporting Information files.

## Abstract

### Background

Agomelatine (AGO) is an antidepressant with unique pharmacological effects; however, its underlying mechanisms remain unknown. In this study, we examined agomelatine's effects on catalase activity, oxidative stress, and inflammation.

### Methods

Chronic restraint stress (CRS) model mice were established over 4 weeks, and AGO 50 mg/kg was administered to different groups alongside a deferasirox (DFX) 10 mg/kg gavage treatment. Behavioral tests were performed to assess the effect of AGO on the remission of depression-like behaviors. Meanwhile, the expression of CAT, the oxidative stress signaling pathway and inflammatory protein markers were assessed using ELISA, qRT-PCR, Western blot, and immunohistochemistry.

### Results

Four weeks of AGO treatment significantly improved depression-like behavior in mice through the activation of catalase in the hippocampus and serum of the model mice, increased superoxide dismutase expression, reduced malondialdehyde expression, and reduced oxidative stress damage. Deferasirox was found to offset this therapeutic effect partially. In addition, the inflammatory pathway (including nuclear factor-κB and nuclear factor of kappa light polypeptide gene enhancer in B cells inhibitor, alpha) was not significantly altered.

**Funding:** This work was supported by The National Natural Science Foundation of China (grant number 8177051246).The funders had no role in study design, data collection and analysis, decision to publish, or preparation of the manuscript.

**Competing interests:** The authors have declared that no competing interests exist.

## Conclusions

AGO can exert antidepressant effects by altering oxidative stress by modulating catalase activity.

## Background

Depression (major depressive disorder, MDD) is one of the most common clinical psychiatric disorders, affecting nearly 350 million people worldwide [1]. Epidemiological surveys have shown that the lifetime prevalence of mood disorders has reached 7.37% [2], with depression accounting for 4.7% [3]. However, because of the enormous disease burden of MDD, the current scientific understanding of its occurrence and progression remains inadequate.

Currently, most antidepressants in clinical use, such as selective serotonin reuptake inhibitors, work by selectively inhibiting serotonin reuptake (5-hydroxytryptamine, 5-HT) in the brain. Agomelatine (AGO), an alternative antidepressant, may exert its anxiolytic effects through an agonist effect on MT1/MT2 receptors and an antagonist effect on 5-HT$_{2C}$ receptors. Lu et al. [4] showed that AGO could improve depression symptoms in mice with chronic unpredictable mild stress (CUMS) by regulating the level of a brain-derived neurotrophic factor in the hippocampus. Previous studies have shown that the antidepressant effects of AGO are not entirely mediated through the same mechanisms as other conventional antidepressants [5–9].

Oxidative stress is an important factor in MDD pathogenesis [10] and is considered one of the main causes of MDD [11]. Maes et al. [12] found that plasma peroxide levels in MDD patients significantly increased compared to normal controls. Moreover, the peroxide levels were significantly higher in patients in the acute phase of the disease than in those with chronic depression (defined as depression lasting more than 2 years). As an important site of reactive oxygen species (ROS) production, the peroxisome plays a crucial role in maintaining the dynamic redox balance in cells and participates in various oxidative stress processes [13]. Catalase (CAT), such as superoxide dismutase (SOD) and glutathione peroxidase, are antioxidative enzymes that can overcome the role of oxidative stress [14]. CAT comprises approximately 40% of all peroxisomal enzymes and is involved in cell proliferation-associated transduction pathways, apoptosis, carbohydrate metabolism, and platelet activation [15]. CAT acts as an antioxidant enzyme by removing excess intracellular hydrogen peroxide ($H_2O_2$) and ROS [16]. Moreover, CAT activity can be significantly suppressed under oxidative stress conditions [17]. Ding et al. [18] conducted an *in vivo* study of depression using constructed fluorescent probe techniques in mouse models. They revealed the excess of peroxisomes and intracellular $H_2O_2$ in the mouse brain, leading to CAT inactivation, dysfunction of the 5-HT system, and, ultimately, depression-related behaviors in mice.

Studies have shown that cysteine and vitamin C can enhance CAT activity and remove excess $H_2O_2$ [19]. In contrast, deferasirox (DFX) can inhibit CAT activity and improve $H_2O_2$ levels [18]. The results of several clinical studies identified elevated CAT activity in the acute and chronic phases of MDD [20–22]. Increased CAT activity may reflect a compensatory mechanism reducing the effects of oxidative stress. However, some studies have presented the opposite conclusion. Ozcan et al. [23] found significantly lower CAT activity in depressed patients than in healthy controls, with no significant change in CAT activity after treatment. However, their study did not describe a specific drug treatment regimen. Bhatt et al. [24] found decreased CAT levels in the brains of mice with CUMS. Olsen et al. [25] found that

antidepressants improved memory and reduced depression and anxiety symptoms by increasing CAT levels in mouse brain tissue, even without increased oxidative stress.

The activation of the oxidative stress pathway is an important pathophysiological factor in depression, based on the proposed hypothesis of the involvement of oxidative stress in the pathogenesis of MDD. AGO may improve depression symptoms, anxiety symptoms, and biochemical indicators of depression by affecting oxidative stress and inflammation.

This study mainly examined the changes in CAT levels and oxidative stress in the brain tissue during AGO treatment in a depression mouse model, determined the presence or absence of a correlation between this change and depression and anxious behavior in mice, and investigated whether this change would be affected by DFX. It was found that AGO can modulate oxidative stress in depression model mice by regulating CAT, thus highlighting new ideas for treating depression.

## Results

### AGO attenuated depressive behavior in depression model mice

The effect of AGO on depression-like behavior was investigated, first analyzing baseline body weight in the six groups of mice (Fig 1A). Secondly, body weight after 28 days of chronic restraint stress (CRS) and drug intervention (Fig 1B) and body weight gain values in the same groups of mice (Fig 1C). Statistical analysis showed that the CRS group lost significantly more body weight than the control group. AGO and AGO + DFX treatment partially offset this weight loss. The therapeutic effect of AGO was partially weakened by DFX, while the DFX treatment group and CRS group did not show any significant differences. At the end of the chronic stress period, a series of well-established behavioral tests that were designed to measure anxiety-like Open Field Test (OFT) and depression-like Sugar-water preference test (SPT), and Forced swimming test (FST) behaviors were conducted in different groups of mice. CRS mice showed lower sucrose preference (Fig 1D), significantly prolonged rest in the FST (Fig 1E), and a significant decrease in total travel in the central region of the OFT (Fig 1F) compared with control mice. These findings were significantly recovered in mice treated with AGO and AGO + DFX compared with those in CRS mice. However, the DFX treatment and CRS groups did not show significant differences in the tests mentioned above. Unfortunately, no significant effects were observed in the total travel time and central region movement time in the OFT tests (Fig 1G–1H). These experiments demonstrated that AGO treatment could restore depressive-like behavior after CRS. However, the therapeutic effect of AGO would be partially attenuated by DFX, indicating that AGO partially improves depressive-like behavior in mice.

### Effect of AGO on CAT levels in mice

The hippocampal sections were immunohistochemically stained with CAT as a specific marker to investigate the effect of AGO on CAT protein expression in the hippocampus of the model mice (Fig 2A). The expression of CAT protein in the CRS group decreased significantly relative to that in the normal control group, and CAT protein expression in the AGO treatment group was significantly restored compared with that in the CRS group (Fig 2B).

Meanwhile, we detected CAT activity in mouse serum (Fig 3A) and found that CAT activity was significantly lower in the CRS group than in the control group. CAT activity increased significantly after AGO treatment, and the DFX had a clear inhibitory effect.

In addition, analysis of CAT protein expression levels by protein immunoblotting (Fig 3D) confirmed that CAT protein expression was significantly lower in the CRS group than in the

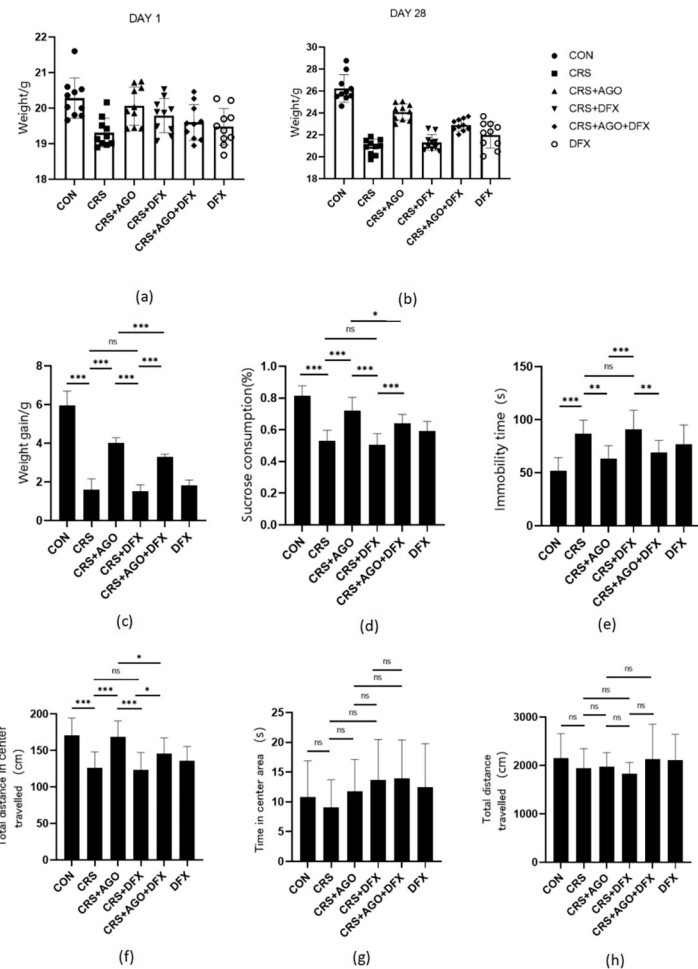

**Fig 1. Effects of the CRS, AGO, and DFX intervention on body weight and behavioral tests.** Depression-like behavior in mice induced by CRS. (a) Initial body weight of the mice. (b) Body weight of the mice after different treatments. (c) Weight gain of the mice. (d) Sucrose preference rate in mice. (e) Forced swimming rest time in mice. (f) The total distance of mice in the central region of the open field test. (g) Total time of mice in the central region in the open field test. (h) The total distance of mice in the open field test. *P<0.05; ** P<0.01; *** P<0.001; CRS, chronic restraint stress; AGO, agomelatine; DFX, deferasirox; ns, no significant difference. Data are expressed as the mean ± standard deviation and were analyzed by one-way ANOVA, followed by post hoc multiple comparisons (LSD method). n = 10.

control group and that AGO treatment effectively restored CAT protein expression (Fig 3E and 3F).

We observed differences in CAT expression by measuring the mRNA expression of CAT in each group through quantitative real-time polymerase chain reaction (PCR), although these were not statistically significant.

In conclusion, we showed that AGO could effectively increase the CAT expression in depressed model mice with CRS.

## Effects of AGO on oxidative stress in depression model mice

The serum concentrations of SOD and malondialdehyde (MDA) were measured using enzyme-linked immunosorbent assay (ELISA) to confirm the effect of AGO on oxidative stress in depression model mice (Fig 3B and 3C). The results showed that the CRS group had

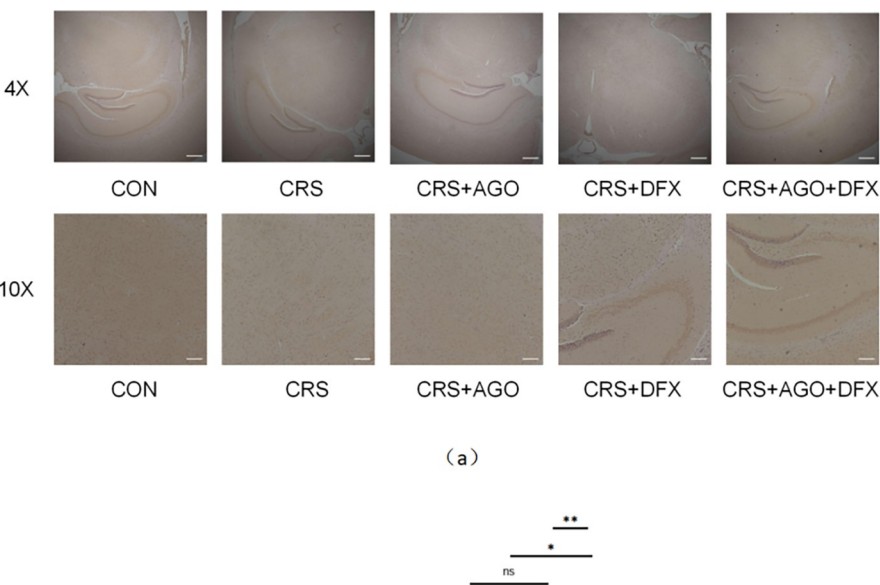

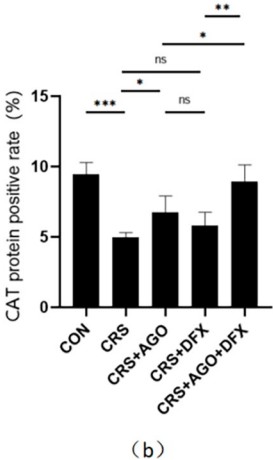

**Fig 2. Effects of the CRS, AGO, and DFX intervention on immunohistochemistry of CAT protein in the hippocampus of mice.** Immunohistochemistry shows the changes in CAT protein expression in the hippocampus of mice after CRS, AGO, and DFX intervention. (a) Staining of CAT protein. (B) Positive rate of CAT protein expression in the mouse hippocampus. *P<0.05; **P<0.01; ***P<0.001; CAT, catalase; CRS, chronic restraint stress; AGO, agomelatine; DFX, deferasirox; ns, no significant difference. Data are expressed as the mean ± standard deviation and were analyzed by one-way ANOVA, followed by post hoc multiple comparisons (LSD method). n = 3.

significantly lower serum MDA concentrations after AGO treatment, while serum SOD increased significantly with AGO treatment. Meanwhile, this therapeutic effect was partially offset by DFX.

## Effect of AGO treatment on the NF-κB pathway in mouse models of depression

Extensive preclinical and clinical evidence shows that external stress is accompanied by increased proinflammatory cytokines and their downstream regulators. Nuclear factor (NF)-κB is a transcription factor reported to regulate an excess of proinflammatory cytokines. The expression levels of the NF-κB protein and IκB-α were analyzed by protein immunoblot (Fig 3G and 3H). Meanwhile, we detected the mRNA expression levels of NF-κB and IκBα by quantitative and real-time PCR (Fig 3I–3K). Unfortunately, our experiments did not reveal significant differences in the expressions of NF-κB and IκBα in each group.

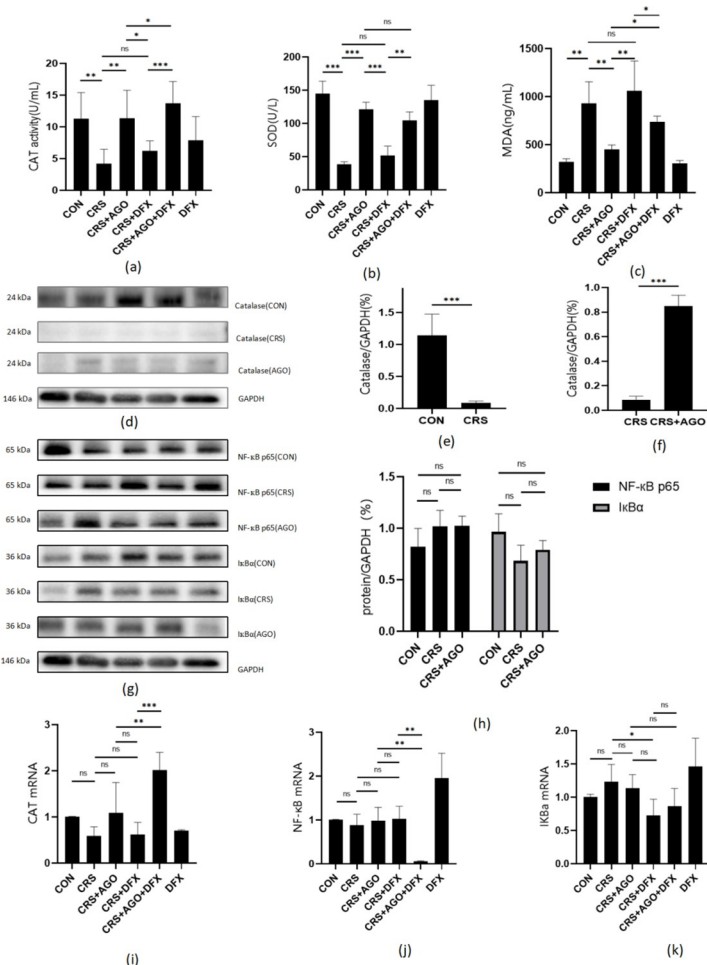

**Fig 3. ELISA, Western blot, and PCR results.** Effect of agomelatine on oxidative stress and the NF-κB pathway in CRS mice. (a) CAT activity in the serum of mice under CRS, AGO, and DFX intervention. (b-c) SOD and MDA concentrations in the serum of mice under CRS, AGO, and DFX intervention. (d, g) Representative protein blot results of CAT, NF-κB p65, and IκBα in the mouse hippocampus under CRS and AGO intervention, showing the quantitative density analysis of CAT in the hippocampus (e, f, h). (i-k) mRNA expression levels of CAT, NF-κB, and IκBα in the mouse hippocampus. *P<0.05; **P<0.01; ***P<0.001; CAT, catalase; CRS, chronic restraint stress; AGO, agomelatine; DFX, deferasirox; SOD, superoxide dismutase; MDA, malondialdehyde; ns, no significant difference. Data are expressed as the mean ± standard deviation and were analyzed by one-way ANOVA, followed by post hoc multiple comparisons (LSD method). n = 3–5.

## Discussion

We hypothesized that the effect of AGO in treating depression is associated with reducing the oxidative stress process in vivo, given the role of oxidative stress in the pathogenesis of depression. CRS induced depression-like behavior in mice, leading to increased MDA and decreased SOD and CAT levels, showing a redox imbalance and oxidative stress state in mice, consistent with previous studies [26]. The marked improvement in depression-like behavior and decreased MDA in depression model mice treated with AGO may be related to increased antioxidant enzymes. In this study, we found that AGO treatment for 4 weeks could activate CAT in the hippocampus of CRS mice, while the CAT inhibitor DFX could partially counteract this therapeutic effect. These findings indicate that AGO can alleviate depression-like behavior in mice by reducing oxidative stress. In addition, inflammation (as determined by NF-κB and

IκBα levels) was not significantly altered, meaning that regulating oxidative stress may be another mechanism of action for depression of AGO [27].

The brain uses aerobic respiration to meet its high energy demand. Although the brain is only 2% of the body's body weight, it consumes approximately 20% oxygen and 25% glucose [28]. Therefore, the central nervous system is very susceptible to oxidative stress, high levels of which can eventually cause oxidative damage, which is mainly manifested as an imbalance of pro-oxidants and antioxidants [29]. SOD converts superoxide anion radicals into $H_2O_2$; CAT, which reduces hydrogen to $H_2O_2$; and glutathione peroxidase are the main enzymatic antioxidants [30]. Previously, it was shown that oxidative stress plays a critical role in the pathophysiology of MDD [31]. Patients with MDD had a lower total antioxidant status and higher MDA levels in their blood than the controls [32]. However, the initially reduced SOD activity and elevated MDA levels in the serum of patients with MDD have been shown to normalize after long-term antidepressant treatment [33,34]. Oxidative stress is an imbalance between ROS and antioxidant enzymes that may cause biomolecular damage. Excess ROS eventually leads to the formation of proinflammatory factors, such as MDA and 8-hydroxy-2-deoxyguanosine, promoting the damage-related molecular patterns of immune responses [35]. MDA inhibits nucleotide excision repair systems and sensitization mutagenesis; disrupts DNA; and causes mitochondrial damage by increasing mitochondrial ROS, inhibiting mitochondrial respiration processes, and reducing mitochondrial membrane potential and antioxidant levels [36]. Superoxide production is increased in the brains of rats with CUMS [37], alongside increased protein and lipid peroxidation and unbalanced SOD and CAT activity [38]. Studies show that elevated oxidative stress has also been found in the brains of adult rats after maternal care deprivation [39]. Our data showed that CRS led to a decreased sugar water preference rate, prolonged forced swimming rest time, and depression-like behaviors such as decreased central total distance in the open field in mice. Meanwhile, CRS also leads to increased MDA expression in the serum, decreased CAT activity in the mouse hippocampus, and decreased SOD expression in the serum. The results of this study are consistent with those of previous studies.

AGO acts as a potent agonist of melatoninergic MT1 and MT2 receptors and as a 5-HT$_{2C}$ receptor antagonist. Evidence accumulated over the extensive course of a previous study [40,41] supports the idea that the antidepressant effect of AGO is because of its synergistic melatoninergic and serotonergic effects [27]. The antidepressant effects of AGO have been illustrated in several mature rodent models reflecting the core clinical features of depression [27]. Chronic (3-week) AGO (40 mg/kg, intraperitoneal injection) has been shown to increase cell proliferation, neurogenesis, and cell survival in the rat hippocampus [39]. Furthermore, AGO improves cell maturity and survival throughout the hippocampus [42]. It has been reported that AGO increases SOD activity in the rat striatum and posterior cortex at 10 mg/kg and 30 mg/kg, respectively [43]. Our results showed that AGO significantly attenuated depressive-like behavior in CRS mice; by increasing CAT activity in the hippocampus and serum, it increased SOD and decreased MDA expression, thus reducing oxidative stress. Meanwhile, we administered the CAT-specific inhibitor DFX to the CRS model mice. We observed that the depression-like behavior and oxidative stress levels in this group were not significantly different from those in the CRS model group. It may be that CAT inhibition has participated in the pathological mechanism of the CRS model. In addition, we concurrently administered DFX and AGO treatment, which showed that the depression-like behavior and oxidative stress levels in the mice were significantly different from those in the AGO treatment group and the CRS group alone. This finding indicated that the therapeutic effect of AGO was produced through multiple different mechanisms and was partly achieved by enhancing CAT expression and activity, thus reducing oxidative stress.

The NF-κB protein family plays an important role in the expression of several proinflammatory genes as a key transcription factor. Regulation of this pathway by bioactive substances may be a potential solution for treating inflammatory diseases [44]. NF-κB is a key mediator of chronic stress-induced depression-like behaviors [45]. Moreover, it has been shown that cellular damage can be induced by the activation of the NF-κB signaling pathway, which is heavily involved in inflammation through ROS and proinflammatory cytokines. Research shows that AGO blocked this activation and the above effects, showing that antidepressant treatment may have broad neuroprotective effects [46]. AGO treatment has been shown to significantly inhibit NF-κB phosphorylation without changing the basal levels of the NF-κB/p65 protein. Because of the inhibition of NF-κB phosphorylation, NF-κ cannot be activated and transported into the nucleus, thus preventing cytokine synthesis [47]. Our experiments showed that NF-κB expression varied within different groups. However, there were no significant differences between groups, possibly because AGO affects NF-κB phosphorylation rather than the basal amount of NF-κB expression.

## Conclusions

In conclusion, our study demonstrated that AGO has a clear antidepressant-like effect in a CRS model, consistent with previous studies. AGO can also reduce oxidative stress in mice by activating CAT, which we believe is one of the underlying antidepressant mechanisms.

## Methods

### Study animals

This study used 6–8-week-old specific pathogen-free male C57BL/6 mice weighing 22±3 g provided by Beijing Weitong Lihua Experimental Animal Company (experimental animal production license no. SCXK [Beijing] 2019–0011) and raised by the Experimental Animal Center of Zhejiang University of Traditional Chinese Medicine (experimental animal use license no. SYXK [Zhejiang] 2019–0022). The animals were adaptively maintained for 1 week. The mice were given a standard pelleted diet and free water, and they were housed at a controlled temperature (23˚C±2˚C) with 60% relative humidity and a 12 h light/dark cycle.

### Model construction and grouping

Establishment of the CRS model [47,48]: the mice were placed in a transparent cylindrical tube with a length of approximately 10 cm and an internal diameter of approximately 3 cm. Therefore, this restricted the head and limb movement so that the head could only be moved slightly. Around the bound tube were eight ventilated holes approximately 0.6 cm in diameter for the mice to breathe. The binding time was from 10:00 to 16:00 (6 h). Each mouse was fed water ad libitum, although the water was not given during the binding process. We have considered the following aspects to ensure that the CRS model has been successfully built as much as possible: The weight of the mice after CRS modeling decreased significantly, with statistical significance.Through behavioral tests, it proved that mice after CRS modeling had depressive behavior. In this experiment, 1) the decreased sugar water preference rate in the sugar-water preference test, 2) the prolonged forced swimming rest time in the forced swimming test, 3) the decreased central total distance in the open field tests.[49,50] Successful mice were studied after 4 weeks of CRS molding.

Group: Sixty mice were randomly divided into six groups of 10 mice each: normal control (CON), CAT inhibitor (DFX), model (CRS), model + AGO (CON + AGO), model + CAT

inhibitor (deferasirox) (CRS + DFX), and model + AGO + CAT inhibitor (CRS + AGO + DFX).

Drug treatment: Animals were treated for 4 weeks through intragastric administration. CRS + AGO mice, CRS + DFX mice, and CRS + AGO + DFX mice were treated with AGO (Jiangsu Hausen Pharmaceutical Company) 50 mg/kg, DFX, or both (Shanghai Bide Medical Technology Company) 10 mg/kg once a day, all drugs were dissolved in normal saline and administered as described in previous studies [5,51]. Mice from the CRS group were given equal normal saline from the first day until the fourth weekend. Mice in the CON and DFX groups were not treated with CRS stimulation. The CON group was administered equal amounts of normal saline (10 mg/kg) once a day; meanwhile, the DFX group was administered DFX (10 mg/kg) once a day.

## Behavior index detection

**Sugar-water preference test (SPT).** This Tang et al. [52] test is divided into adaptation and experimental stages. Adaptation stage: Single-cage feeding, with two water bottles placed simultaneously. Both bottles were filled with 1% sucrose water Within the first 24 h. One bottle contained pure water for the subsequent 24 h, and the other contained 1% sucrose water. Experimental stage: one bottle of 1% sucrose water and one bottle of pure water were given after 24 h of the SPT, and the overall weights of the bottles were recorded. The positions of the two water bottles were exchanged after 12 h, and their overall weight was re-recorded after 24 h. The sugar water preference rate was calculated as the indicator of the SPT (sugar water preference rate = sugar water consumption/[sugar water consumption + pure water consumption] × 100%).

**Open Field Test (OFT).** SMART 3.0 virtually divided the absent field analysis box (40 cm × 40 cm × 40 cm) into central and peripheral parts using the Song et al. [53] OFT. Experiments were performed in a darker observation box. First, the mice were placed in the central grid to be observed for 5 min. After the experiment, the field box was wiped to remove the residual odor, and the next animal was tested. The main observation indicators were the central crossing time (s) and the total distance (cm).

**Forced swimming test (FST).** Using the Lin et al. [54] FST, mice were placed in a circular transparent swimming bucket, 30 cm high, 12 cm in diameter, and 18 cm deep (i.e., that the tail and hind limbs could not touch the bottom), with a water temperature of 25°C±1°C. Each mouse first swam for 1 min to adapt to the environment. The behavioral indicators of the mice were recorded with a camera for 5 min, and their stationary time was recorded. The mice were dried with a towel after the experiment.

## Immunohistochemistry

Mice were perfused with 4% paraformaldehyde (PFA) after being perfused with 50 mL (±) of saline through the heart for the immunohistochemistry study, and their brains were removed. The hippocampus was isolated and fixed in 4% paraformaldehyde for 3 days at 4 °C after that. Fixed Hippocampal tissue was embedded in paraffin and sectioned to 10μm thickness. Paraffin tissue sections were taken and baked at 60°C for 1 h; dried in three cylinders of xylene for 10 min; dehydrated by 95%, 80%, and 75% gradient alcohol for 1 min; immersed in tap water for 1 min; soaked in 3% $H_2O_2$ at 37°C for 30 min, phosphate-buffered saline (PBS) for 3 min, and 0.01 M citrate buffer for 20 min at 95°C; cooled to room temperature; and washed with PBS. Normal sheep serum working fluid was blocked at 37°C for 10 min. One-drop antibody plus rabbit anti-rat CAT (ab209211, Abcam) was added at 4°C overnight and rinsed in PBS. It was incubated with horseradish-labeled secondary antibody for 30 min in DAB. The section was

re-stained with hematoxylin and sealed. PBS was used instead of the primary antibody as a negative control, and the normal mucosa was used as a positive control. The positive cell rate was calculated by randomly selecting five high-power fields (400X) and counting 100 cells per field [5,55].

## ELISA experiments

The mice were anesthetized with 0.3% (0.15 ml/10 g) pentobarbital sodium by intraperitoneal injection. Blood was collected through a puncture of the retrobulbar venous plexus after anesthesia was deemed effective. Approximately 0.2 ml of blood was collected in a labeled centrifuge tube and kept at 20°C for 1–2 h. The blood was precooled to 4°C and centrifuged at 3000 rpm/min for 10 min. The serum was pipetted into a newly labeled centrifuge tube, frozen, and stored at -80°C for backup. Mouse catalase (CAT) ELISA Kit (ml037752, Enzyme-Linked Biotechnology), mouse superoxide dismutase (SOD) ELISA kit (ml643059, Enzyme-Linked Biotechnology), and MDA (Malondialdehyde) ELISA Kit (E-EL-0060c, Elabscience) content were determined according to the manufacturer's instructions.

## CAT activity detection

The methodology was carried out according to the manufacturer's instructions (BC0200, Solarbio) [56–58]. Ice bath homogenate at a ratio of tissue mass (g): extract volume (mL) of 1:5°C–10.4°C was centrifuged at 8000 g for 10 min. The supernatant was then removed and placed on ice for testing. The working fluid was then configured according to the instructions. The spectrophotometer was preheated for more than 30 min, the wavelength was adjusted to 240 nm, and the distilled water was adjusted to 0. The CAT working fluid was detected for 10 min before determination in a 25°C water bath. Then, 1 mL of CAT detection solution was added to a 1 mL quartz plate before 35 L samples were added and mixed for 5 s. The initial light absorption at 240 nm and light absorption after 1 min were measured immediately at room temperature.

## Quantitative real-time-PCR

Primers for CAT, GAPDH, NF-κB, and IκBα were designed and submitted to TaKaRa for synthesis. Total RNA was extracted with the RNA extraction kit. The reverse transcription system was 20 μL, which was performed according to the reagent instructions. The reaction solution was taken for quantitative PCR according to the kit's instructions. GAPDH was used as the internal reference control, and 2-ΔΔCt calculated the relative expression content of the target gene expression. Each group of samples was tested more than three times.

## Western blot

Each group of mice was killed by cervical dislocation after completing the last behavioral test. Each subject's brain was dissected and quickly placed on the ice surface. The hippocampal

| Gene | Forward primer (5'-3') | Reverse primer (5'-3') |
|---|---|---|
| NF-κB [59] | CCGCTCGAGCTATGGACGATCTGTTTCCCCTC | CGGAATTCACCTTAGGAGCTGATCTGAC |
| CAT [60] | GCAGATACCTGTGAACTGTC | GTAGAATGTCCGCACCTGAG |
| IκBα [61] | CACTCCATCCTGAAGGCTACCAA | AAGGGCAGTCCGGCCATTA |
| GAPDH [62] | TGTGGGCATCAATGGATTTGG | ACACCATGTATTCCGGGTCAAT |

tissue was quickly isolated according to the stereotactic map of the mouse brain, immediately placed in liquid nitrogen, and then moved to an -80˚C refrigerator for storage. The brain tissue was ground to a homogenate for the subsequent experimental testing, and a radioimmunoprecipitation assay lysate was added on ice for 30 min. The solution was centrifuged at 4˚C at 14400 rpm for 15 min. The supernatant was removed and quantified with a bicinchoninic acid assay kit using a 5-protein loading buffer at 95˚C for 10 min. SDS-PAGE electrophoresis was used to separate the protein, transferred to a polyvinylidene fluoride membrane, and then blocked for 1 h at room temperature with 5% skim milk powder. The membrane was then washed three times in TBST plus CAT (ab209211, Abcam), NF-κB (ab207297, Abcam), IκBα (ab32518, Abcam), and GAPDH (ab9484, Abcam). The primary antibody was incubated overnight at 4˚C and then washed away with TBST. Then the corresponding secondary antibody was added for 1 h. The membrane was washed in TBST three times and swept with the ODYSSEY two-color infrared laser imaging system. The experimental results were processed with Image Studio Ver 2.0 software.

### Statistical analysis

SPSS 25.0 software (IBM Corp., Armonk, NY, USA) was used for the statistical analysis. All data are presented as mean ± standard deviation. Multi-group overall comparisons were analyzed using one-way ANOVA and post hoc multiple comparisons (LSD method). A P-value of <0.05 was considered statistically significant.

### Supporting information

**S1 File.**
(ZIP)

### Acknowledgments

We appreciate the Experimental Animal Center of Zhejiang Chinese Medical University for the technical assistance they provided.

### Author Contributions

**Conceptualization:** Cheng Zhu.

**Data curation:** Jiaxi Xu.

**Formal analysis:** Jiaxi Xu.

**Funding acquisition:** Enyan Yu.

**Investigation:** Wangdi Sun.

**Methodology:** Cheng Zhu.

**Validation:** Piaopiao Jin.

**Writing – original draft:** Jiaxi Xu.

**Writing – review & editing:** Enyan Yu.

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
