## [Decision Letter · Decision Letter 0]

8 Mar 2023

PONE-D-23-03416Agomelatine improves depression through mechanisms affecting oxidative stress and inflammation via altered catalase activityPLOS ONE

Dear Dr. Xu,

Thank you for submitting your manuscript to PLOS ONE. After careful consideration, we feel that it has merit but does not fully meet PLOS ONE’s publication criteria as it currently stands. Therefore, we invite you to submit a revised version of the manuscript that addresses the points raised during the review process.

We look forward to receiving your revised manuscript.

Kind regards,

Mohamed Abdel-Daim, Ph.D.

Academic Editor

PLOS ONE

Journal Requirements:

"This work was supported by The National Natural Science Foundation of China (grant number 8177051246)."

5. Thank you for stating the following in the Funding Section of your manuscript: 

"This work was supported by The National Natural Science Foundation of China (grant number 8177051246)."

"This work was supported by The National Natural Science Foundation of China (grant number 8177051246)."

Reviewers' comments:

Reviewer's Responses to Questions

**Comments to the Author**

1. Is the manuscript technically sound, and do the data support the conclusions?

Reviewer #1: Partly

Reviewer #2: Yes

2. Has the statistical analysis been performed appropriately and rigorously? 

Reviewer #1: No

Reviewer #2: Yes

3. Have the authors made all data underlying the findings in their manuscript fully available?

Reviewer #1: Yes

Reviewer #2: Yes

4. Is the manuscript presented in an intelligible fashion and written in standard English?

Reviewer #1: Yes

Reviewer #2: Yes

5. Review Comments to the Author

Reviewer #1: The current work aimed to investigate the possible antidepressant effect of agomelatine (AGO) and the role of the catalase (CAT) enzyme in mediating this effect. In addition, the theories of increased oxidative stress and neuroinflammation were tested in chronic restraint stress model. The manuscript follows the standard scientific way; however, it needs much improvement. I have The following concerns about the manuscript:

1) Title

It doesn’t express the results of the current study in a good way because 1) AGO was given from day 1 of the study and not after the development of the chronic restraint stress (CRS) model; 2) There were no significant differences regarding the inflammatory markers tested between all groups; 3) Blocking catalase by deferasirox didn’t worsen the CRS model. The authors could think of another title, I suggest “Agomelatine prevented depression in chronic restraint stress model through enhanced catalase activity and halted oxidative stress”. If significant results were found regarding the inflammatory markers after revising the post hoc analysis of ANOVA, you may express this in the title as well.

2) Abstract

• Page 2, Line 21 (first line): The abbreviation of agomelatine (AGO) should be mentioned and used subsequently after that, see also line 32 & 39.

3) Introduction

• Page 3, lines 52, 53: Please, modify the sentence “agomelatine (AGO) may exert its anxiolytic, depressive effects …” and choose a better word than “depressive” to express the drug-calming effect; otherwise, remove it. This is because “depressive” is not logical while the study aimed to test agomelatine in the treatment of depression.

• Page 3, line 56: Please, add references to the sentence stating “Previous studies”.

• Page 4, line 67: “glutathione” is not an antioxidant enzyme, please edit to “glutathione peroxidase”.

• Page 4, line 67: The word “neutralize” is better to be changed to “overcome”.

• Page 5, line 89: The Phrase “we propose a novel hypothesis” is not accurate since the relation between depression and increased ROS has been proposed many years ago. It is better to be edited to “Based on the proposed hypothesis of the involvement of oxidative stress in the pathogenesis of MDD, …”

• Page 5, line 90: Please, edit the phrase “AGO can improve depression” to “AGO may improve depression” so that the research question carries the possibility of the desired effect.

4) Results

• Page 6, line 117: Please, edit the “abovementioned trials” by adding a space after the word “above” and replacing “trials” with “tests”. Also, replace the word “trial” in line 118.

• Page 6, line 122: The phrase “through routes that enhance CAT activity” is not appropriate in place because, in this paragraph, you only express the behavioral test results and the results of oxidative stress have not been mentioned yet.

• Page 8, line 152: Please, delete “CAT inhibitor” because deferasirox is primarily classified as an iron chelator, and in the results, we only mention the effect of the tested drugs without explanation.

• Page 9, line 177: Please, edit this sentence “The results showed that the CRS group had significantly lower serum SOD and MDA concentrations after AGO treatment” as serum SOD was significantly increased with AGO treatment.

• Page 9, line 179: Please, remove this sentence “This is because CRS causes elevated levels of oxidative stress, and AGO was shown to attenuate oxidative stress in mouse models of depression, potentially because of its effects on CAT activity” as it is an explanatory sentence. It should be placed within the discussion.

• Page 9, line 185: NF-κB is a downstream regulator of inflammatory cytokines. Please, correct the word “upstream regulators”.

5) Discussion

• Page 10, line 203: This sentence is vague “This means that, except in melatonin and 5-HT2C, the synergistic effect of regulating oxidative stress may be another mode of action for AGO”. Please, edit the sentence to be more clear and express the desired aim.

• Page 11, line 228: The sentence “In turn, this leads to increased CAT expression…” should be edited and clarified whether you want to refer to the results of the CRS or AGO results. Also, revise the changes of CAT, SOD, and MDA accordingly.

• Page 12, line 246: The following inference is not correct “demonstrating that oxidative stress may be caused by the inhibition of CAT activity”. It refers that other mechanisms than CAT inhibition participating in the CRS model &/or MDD pathogenesis. Your inference could be correct if the administration of DFX further worse the CRS model. Please, explain or criticize.

• Page 13, line 259: Please, edit the sentence starting with “AGO blocked this activation …” to show that these are the result of a previous study (reference 39) and not the current study.

6) Conclusion

• Page 13, line 270: Please, edit the phrase “is the underlying antidepressant mechanism” to “is one of the underlying antidepressant mechanisms”.

7) Methods

• Page 13, line 272: “Study subjects” is better to be replaced with “Study animals”.

• Page 14 (Model construction and grouping): Please, add references to the paragraph on model establishing.

• Page 14, line 285: Please, check this phrase for grammar “Each mouse was fed water on an ad libitum diet”. I think the preposition should be removed “Each mouse was fed water ad libitum”.

• Page 14, line 286: Please, mention the criteria of “Successful mice” and a reference for that.

• Page 14, line 290: Please, correct the drug name “dilarosine” to “deferasirox”.

• Page 14, lines 291-297: Please, mention the references for the selected dose of the drugs. Also, mention the route of administration clearly and edit “saline infusion for gavage”. How did you calculate the dose of the saline (normal saline 10 mg/kg/day) while it is a solution?

• Page 15: Please, add references to the Sugar-water preference test, Open Field Test, and Forced swimming test.

• Page 16: Please, add references to Immunohistochemistry. Also, specify which part of the brain has been used in paraffin tissue sections.

• Page 17, line 340: Please, mention the catalog number and the scientific name of the ELISA kits used to assess CAT, SOD, and MDA.

• Page 17, line 342: Please, add the reference for CAT activity detection.

• Page 17, line 354: Please, mention the reference for the primers’ design.

• Page 18, line 381: It is illogical to use the T-test to compare between two groups' mean since SPSS contains post hoc tests after ANOVA. Please, recheck all your statistical analysis with the appropriate post hoc test again.

8) Figure legends

• Figure 1 legend (Page 6, line 124:) I suggest using the following title “Effects of the CRS, AGO, and DFX intervention on body weight and behavioral tests” and remove the phrase “Effects of the CRS, AGO, and DFX intervention” from the subsequent subtitle. Please, edit as appropriate and add the name of the statistical test used.

• Figure 2 legend (Page 7, line 143): Please, edit the title to “Effects of the CRS, AGO, and DFX intervention on immunohistochemistry of CAT protein in the hippocampus of mice” and add the name of the statistical test used.

• Figure 3 legend (Page 8, line 157): Please, add the name of the statistical test used.

9) English proof

• Please, the manuscript has to be checked by a native English speaker.

Thank you and best wishes.

Reviewer #2: The manuscript is discussing the molecular mechanisms underlying the anti-depressant effect of Agomelatine.

some important questions need to be answered:

1-Where the blood samples taken are collected from?

2- What is the reference for the duration of the induction method?

3- How is the doses for the used drugs determined ?

6. PLOS authors have the option to publish the peer review history of their article (what does this mean?). If published, this will include your full peer review and any attached files.

Reviewer #1: No

Reviewer #2: **Yes: **Sherihan SalahEldin Abdelhmaid

---

## [Author Response · Author response to Decision Letter 0]

14 Mar 2023

Thanks for taking your time to review this manuscript. We really appreciate all editors and reviewers detailed comments and suggestions! According to editors valuable comments, we ensure that our manuscript meets PLOS ONE's style requirements. As suggested, we have attached the Funding Statement and Role of Funder statement in the resubmission cover letter and the ethics statement is moved to the Methods section in the Manuscript.

According to reviewers valuable comments, we have made corresponding changes and adjustments in the Revised Manuscript with Track Changes and the Response to Reviewers. Those changes are highlighted in blue within the re-submitted files. 

We hope you will find this revised version satisfactory and look forward to hearing from you. We would be glad to respond to any further questions and comments that you may have.

---

## [Decision Letter · Decision Letter 1]

15 May 2023

PONE-D-23-03416R1Agomelatine prevented depression in the chronic restraint stress model through enhanced catalase activity and halted oxidative stressPLOS ONE

Dear Dr. Xu,

Thank you for submitting your manuscript to PLOS ONE. After careful consideration, we feel that it has merit but does not fully meet PLOS ONE’s publication criteria as it currently stands. Therefore, we invite you to submit a revised version of the manuscript that addresses the points raised during the review process.

We look forward to receiving your revised manuscript.

Kind regards,

Mohamed Abdel-Daim, Ph.D.

Academic Editor

PLOS ONE

Journal Requirements:

Reviewers' comments:

Reviewer's Responses to Questions

**Comments to the Author**

1. If the authors have adequately addressed your comments raised in a previous round of review and you feel that this manuscript is now acceptable for publication, you may indicate that here to bypass the “Comments to the Author” section, enter your conflict of interest statement in the “Confidential to Editor” section, and submit your "Accept" recommendation.

Reviewer #1: All comments have been addressed

2. Is the manuscript technically sound, and do the data support the conclusions?

Reviewer #1: Yes

3. Has the statistical analysis been performed appropriately and rigorously? 

Reviewer #1: Yes

4. Have the authors made all data underlying the findings in their manuscript fully available?

Reviewer #1: Yes

5. Is the manuscript presented in an intelligible fashion and written in standard English?

Reviewer #1: Yes

6. Review Comments to the Author

Reviewer #1: Thank you so much. The authors addressed most of my concerns, but minor changes should be considered:

- Page 11, line 230: Please, add the references of the “previous studies” mentioned.

- Page 14, line 286: Please, add the criteria of “Successful mice” with references to the revised manuscript.

- Page 17, line 354: Please, add the sequences of the primers’ and the references to the revised manuscript.

Best wishes.

7. PLOS authors have the option to publish the peer review history of their article (what does this mean?). If published, this will include your full peer review and any attached files.

Reviewer #1: No

---

## [Author Response · Author response to Decision Letter 1]

23 Jun 2023

Response Letter to Reviewers

Thanks very much for taking the time to review this manuscript and providing us with this great opportunity to submit a revised version of our manuscript (Manuscript: PONE-D-23-03416). We appreciate all your detailed comments and suggestions! We have incorporated them, and those changes are highlighted in yellow within the re-submitted files. We hope this revised manuscript has addressed your concerns and look forward to hearing from you.

I have the following replies to the comments of reviewers.

# Comments from reviewer 1:

Thank you so much. The authors addressed most of my concerns, but minor changes should be considered:

- Page 11, line 230: Please, add the references of the “previous studies” mentioned.

- Page 14, line 286: Please, add the criteria of “Successful mice” with references to the revised manuscript.

- Page 17, line 354: Please, add the sequences of the primers’ and the references to the revised manuscript.

Best wishes.

# Authors response:

Thank you very much for your time in reviewing the manuscript and your encouraging comments on the merits. We also appreciate your clear feedback and hope that you will find that the explanation has fully addressed all of your concerns and that this re-submitted manuscript is satisfactory. Below, we will specifically address the comments point-by-point, and changes are highlighted in yellow within the re-submitted files.

Comment 1:” Page 11, line 230: Please, add the references of the “previous studies” mentioned.”

Response:

Thank you for the detailed review to improve our manuscript. As suggested by the reviewer, we add the references of the “previous studies”. (Page 11, lines 231)

“Evidence accumulated over the extensive course of a previous study[40,41] supports the idea that the antidepressant effect of AGO is because of its synergistic melatoninergic and serotonergic effects [29].”

Reference:

40. Millan MJ, Marin P, Kamal M, Jockers R, Chanrion B, Labasque M, et al. The melatonergic agonist and clinically active antidepressant, agomelatine, is a neutral antagonist at 5-HT(2C) receptors. Int J Neuropsychopharmacol. 2011;14(6):768-83. doi: 10.1017/s1461145710001045.

41. Norman TR, Cranston I, Irons JA, Gabriel C, Dekeyne A, Millan MJ, et al. Agomelatine suppresses locomotor hyperactivity in olfactory bulbectomised rats: a comparison to melatonin and to the 5-HT(2c) antagonist, S32006. European journal of pharmacology. 2012;674(1):27-32. doi: 10.1016/j.ejphar.2011.10.010.

Comment 2: ”Page 14, line 286: Please, add the criteria of “Successful mice” with references to the revised manuscript.”

Response:

Thank you for the detailed review to improve our manuscript. As suggested by the reviewer, we have added this important information to our re-submitted manuscript. (Page 14, lines 283-289)

“We have considered the following aspects to ensure that the CRS model has been successfully built as much as possible: The weight of the mice after CRS modeling decreased significantly, with statistical significance.Through behavioral tests, it proved that mice after CRS modeling had depressive behavior. In this experiment, 1) the decreased sugar water preference rate in the sugar-water preference test, 2) the prolonged forced swimming rest time in the forced swimming test, 3) the decreased central total distance in the open field tests.[49,50]”

References:

49. Yan HC, Cao X, Das M, Zhu XH, Gao TM. Behavioral animal models of depression. Neurosci Bull. 2010;26(4):327-37. doi: 10.1007/s12264-010-0323-7.

50. Campos AC, Fogaça MV, Aguiar DC, Guimarães FS. Animal models of anxiety disorders and stress. Revista brasileira de psiquiatria (Sao Paulo, Brazil : 1999). 2013;35 Suppl 2:S101-11. doi: 10.1590/1516-4446-2013-1139.

Comment 3:“Page 17, line 354: Please, add the sequences of the primers’ and the references to the revised manuscript.”

Response: 

Thank you for pointing this out. We have revised it in our manuscript to reflect your great suggestion. (Page 18, lines 366)

Gene Forward primer（5’-3’） Reverse primer（5’-3’）

NF-κB[59] CCGCTCGAGCTATGGACGATCTGTTTCCCCTC CGGAATTCACCTTAGGAGCTGATCTGAC

CAT[60] GCAGATACCTGTGAACTGTC GTAGAATGTCCGCACCTGAG

IкBα[61] CACTCCATCCTGAAGGCTACCAA AAGGGCAGTCCGGCCATTA

GAPDH[62] TGTGGGCATCAATGGATTTGG ACACCATGTATTCCGGGTCAAT

References:

59. Yuan YH, Sun JD, Wu MM, Hu JF, Peng SY, Chen NH. Rotenone could activate microglia through nfκb associated pathway. Neurochem Res. 2013;38: 1553-1560. doi: 10.1007/s11064-013-1055-7.

60. El Mouatassim S, Guérin P, Ménézo Y. Expression of genes encoding antioxidant enzymes in human and mouse oocytes during the final stages of maturation. Mol Hum Reprod. 1999;5: 720-725. doi: 10.1093/molehr/5.8.720.

61. Feng X, Wang H, Ye S,  Guan J, Tan W, Cheng S, et al. Up-regulation of microRNA-126 may contribute to pathogenesis of ulcerative colitis via regulating NF-kappaB inhibitor IκBα. PloS One. 2012;7: e52782. doi: 10.1371/journal.pone.0052782.

62. Zha H, Miao W, Rong W, Wang A, Jiang W, Liu R, et al. Remote ischaemic perconditioning reduces the infarct volume and improves the neurological function of acute ischaemic stroke partially through the miR-153-5p/TLR4/p65/IkBa signalling pathway. Folia Neuropathol. 2021;59: 335-349. doi: 10.5114/fn.2021.112127.

Finally, we would like to take this opportunity to thank you again for all your time and for providing us with this excellent opportunity to improve the manuscript. We hope you will find this revised version satisfactory.

---

## [Editor Report · Decision Letter 2]

14 Jul 2023

Agomelatine prevented depression in the chronic restraint stress model through enhanced catalase activity and halted oxidative stress

PONE-D-23-03416R2

Dear Dr. Xu,

We’re pleased to inform you that your manuscript has been judged scientifically suitable for publication and will be formally accepted for publication once it meets all outstanding technical requirements.

Kind regards,

Mohamed Abdel-Daim, Ph.D.

Academic Editor

PLOS ONE
---

## [Editor Report · Acceptance letter]

2 Feb 2024

PONE-D-23-03416R2 

PLOS ONE

Dear Dr. Yu, 

I'm pleased to inform you that your manuscript has been deemed suitable for publication in PLOS ONE. Congratulations! Your manuscript is now being handed over to our production team.

Kind regards, 

on behalf of

Professor Mohamed Abdel-Daim 

Academic Editor

PLOS ONE